# Phosphorylations and Acetylations of Cytochrome *c* Control Mitochondrial Respiration, Mitochondrial Membrane Potential, Energy, ROS, and Apoptosis

**DOI:** 10.3390/cells13060493

**Published:** 2024-03-12

**Authors:** Paul T. Morse, Tasnim Arroum, Junmei Wan, Lucynda Pham, Asmita Vaishnav, Jamie Bell, Lauren Pavelich, Moh H. Malek, Thomas H. Sanderson, Brian F.P. Edwards, Maik Hüttemann

**Affiliations:** 1Center for Molecular Medicine and Genetics, Wayne State University, Detroit, MI 48201, USA; gd0281@wayne.edu (P.T.M.);; 2Department of Biochemistry, Microbiology, and Immunology, Wayne State University, Detroit, MI 48201, USA; 3Division of Pediatric Critical Care, Children’s Hospital of Michigan, Central Michigan University, Detroit, MI 48201, USA; 4Department of Health Care Sciences, Eugene Applebaum College of Pharmacy & Health Sciences, Wayne State University, Detroit, MI 48201, USA; 5Department of Emergency Medicine, University of Michigan Medical School, Ann Arbor, MI 48109, USA

**Keywords:** cytochrome *c*, post-translational modifications, apoptosis, respiration, mitochondria, electron transport chain

## Abstract

Cytochrome *c* (Cyt*c*) has both life-sustaining and cellular death-related functions, depending on subcellular localization. Within mitochondria, Cyt*c* acts as a single electron carrier as part of the electron transport chain (ETC). When released into the cytosol after cellular insult, Cyt*c* triggers the assembly of the apoptosome, committing the cell to intrinsic apoptosis. Due to these dual natures, Cyt*c* requires strong regulation by the cell, including post-translational modifications, such as phosphorylation and acetylation. Six phosphorylation sites and three acetylation sites have been detected on Cyt*c* in vivo. Phosphorylations at T28, S47, Y48, T49, T58, and Y97 tend to be present under basal conditions in a tissue-specific manner. In contrast, the acetylations at K8, K39, and K53 tend to be present in specific pathophysiological conditions. All of the phosphorylation sites and two of the three acetylation sites partially inhibit respiration, which we propose serves to maintain an optimal, intermediate mitochondrial membrane potential (ΔΨ_m_) to minimize reactive oxygen species (ROS) production. Cyt*c* phosphorylations are lost during ischemia, which drives ETC hyperactivity and ΔΨ_m_ hyperpolarization, resulting in exponential ROS production thus causing reperfusion injury following ischemia. One of the acetylation sites, K39, shows a unique behavior in that it is gained during ischemia, stimulating respiration while blocking apoptosis, demonstrating that skeletal muscle, which is particularly resilient to ischemia-reperfusion injury compared to other organs, possesses a different metabolic strategy to handle ischemic stress. The regulation of Cyt*c* by these post-translational modifications underscores the importance of Cyt*c* for the ETC, ΔΨ_m_, ROS production, apoptosis, and the cell as a whole.

## 1. Introduction

In healthy mitochondria, the electron transport chain (ETC) together with ATP synthase produces the majority of cellular ATP via the oxidative phosphorylation (OxPhos) process [1]. Four complexes and two electron carriers form the ETC. Electrons enter through either complex I, via NADH, or complex II, via FADH_2_, which couples the citric acid cycle to the ETC. These electrons are transferred to the ubiquinone pool, generating ubiquinol, which reduces complex III. The electrons then reach cytochrome *c* (Cyt*c*), which passes them to complex IV (cytochrome *c* oxidase, COX). The movement of Cyt*c* between complex III and COX has been proposed to occur through 2D diffusion or restricted diffusion by Cyt*c* sliding between the complexes, making it more efficient compared to the free diffusion model [2,3]. Finally, COX catalyzes the reduction of oxygen, the terminal electron acceptor, to water. Complexes I, III, and IV pump protons from the matrix into the intermembrane space (IMS). This generates the proton-motive force, which is composed of both a pH gradient and the mitochondrial membrane potential (ΔΨ_m_) [4] allowing ATP synthase (complex V) to harnesses this electrochemical gradient to produce ATP.

Naturally, this process is tightly regulated to ensure that ATP production is equivalent to energy demand. In addition to the complexes being regulated at the transcriptional and translational level, other mechanisms, such as supercomplex formation, allosteric binding of ADP and ATP, and post-translational modifications, allow for direct control over complex activity [2,5,6,7]. There is proposed to be crosstalk between these regulatory mechanisms, as post-translational modifications of Cyt*c* may optimize supercomplex functioning [3]. However, these processes can go awry, with the resulting mitochondrial dysfunction contributing to a variety of pathologies, including neurodegeneration, ischemia-reperfusion injury, and cancer [8]. In many disease conditions, mitochondrial dysfunction triggers increased reactive oxygen species (ROS) production, which then initiates the release of Cyt*c* into the cytosol [9]. Cyt*c* release from the mitochondria takes place via a two-step process. First, Cyt*c*, which is typically associated with cardiolipin within the cristae, dissociates from cardiolipin following cardiolipin peroxidation [10,11]. Cristae remodeling and swelling via tBID allows for further Cyt*c* clearance out of the cristae [12]. Second, Cyt*c* must cross the outer mitochondrial membrane, which occurs via calcium-dependent mechanisms such as the permeability transition pore or calcium-independent mechanisms via the Bcl-2 family proteins [13,14,15]. Once released from the mitochondria into the cytosol, Cyt*c* is rapidly reduced in otherwise healthy cells [16]. However, in apoptotic cells, it is the oxidized form that binds to apoptosis protease-activating factor-1 (Apaf-1), leading to formation of the apoptosome and activation of the caspase cascade, because Cyt*c* is rapidly oxidized by COX to which it has access due to outer mitochondrial membrane permeabilization [17,18]. Pharmacological interventions, such as treatment with anthocyanins that reduce cytosolic Cyt*c*, hold promise as tissue protective treatments by inhibiting apoptosome formation [19,20]. In cancer, it is worth noting that sublethal release of Cyt*c* into the cytosol can desensitize the cancer cell to apoptosis [21]. In effect, the subcellular localization and regulation of Cyt*c* controls the balance between cellular life and death. This review discusses how tissue-specific post-translational modifications of Cyt*c* affect the functions of the protein as well as its impact on various disease pathologies. While reviews of post-translational modifications of Cyt*c* exist [22,23], new discoveries since then demand updating the literature.

## 2. Various Functions of Cyt*c*

Cyt*c* is a small, globular, 104 amino acid, evolutionarily optimized protein that carries out a variety of distinct functions. Cyt*c* possesses a covalently attached heme group, which is crucial for carrying out its redox-based functions. The two major functions of Cyt*c* were already introduced above: it operates a single electron carrier from complex III to COX of the mitochondrial ETC and initiates intrinsic apoptosis via binding to Apaf-1 when released into the cytosol. Cyt*c* also supports other crucial, life-sustaining processes (Figure 1). For example, Cyt*c* scavenges and detoxifies ROS including superoxide and hydrogen peroxide [24,25,26] and acts as an electron acceptor of the redox-coupled import pathway of proteins to the IMS via the Erv1-Mia40 pathway [27]. Additionally, a new area of investigation is the role of Cyt*c* in nuclear-mitochondrial crosstalk. In response to DNA damage, Cyt*c* has been shown to translocate to the nucleus, where it promotes both Protein Phosphatase 2A activity via interaction with histone chaperone ANP32A and the release of alternative reading frame protein from nucleophosmin, facilitating DNA damage repair [28,29,30]. However, it still has to be determined if this function of Cyt*c* can rescue cells from apoptosis or only delay it, because binding to Apaf-1 in the cytosol competes with its nuclear function.

Moreover, Cyt*c* is involved in a variety of pro-apoptotic processes beyond its interaction with Apaf-1, and dysfunction in programmed cell death is relevant to numerous disease processes. Cyt*c* is able to produce ROS via p66^Shc^ [31,32], which can damage the cell and trigger cell death when excessive ROS are generated. As an early step during apoptosis, Cyt*c* catalyzes the formation of cardiolipin peroxide [11,33,34,35]. Cardiolipin remodeling is an on-going research topic, which is also related to supercomplex formation, and defects in this process have been implicated in different disease pathologies [36,37,38]. Interestingly, the cardiolipin peroxidase activity of Cyt*c* may be driven by ROS produced during ischemia-reperfusion injury [39].

Overall, the functional diversity of Cyt*c* underscores the need for tight regulatory control, particularly when cellular fate hangs in the balance. Various post-translational modifications of Cyt*c* have been discovered, some of which have been functionally characterized (Figure 2). Some of these modifications, or the loss thereof, result in dysregulation of Cyt*c* and have been associated with specific pathologies. Although other post-translational modifications, such as methylation, nitration, nitrosylation, and homocysteinylation, have been reported, this review focuses on the effects of phosphorylations and acetylations of Cyt*c* that have been identified in vivo, along with any associated pathologies.

## 3. Characterized Phosphorylation Sites of Cyt*c*

Phosphorylation of Cyt*c* (Table 1) has been identified on threonine 28 (T28), serine 47 (S47), tyrosine 48 (Y48), threonine 49 (T49), threonine 58 (T58), and tyrosine 97 (Y97). These residues, except for T58, which is replaced with isoleucine in humans, are conserved in mammals. Cyt*c* was not known to be regulated via cell-signaling mechanisms until the discovery of Y97 phosphorylation in 2006, which was mapped in bovine heart [40]. This discovery was made using protein purification protocols that preserve the in vivo phosphorylation state of mitochondrial proteins. Despite this advance, studying post-translational modifications of Cyt*c* remains challenging. Protein purified from animal tissue exists as a mixture of modified and unmodified forms, and depending on the tissue, yields can be low. Therefore, these phosphorylation sites were also functionally characterized using phosphomimetic replacement strategies, most commonly by glutamate replacement, which mimics the negative charge of the phosphate group. Following Y97 phosphorylation, Y48 phosphorylation was identified in bovine liver [41], followed by T28 [42] and T58 [43] in bovine and rat kidney, respectively, S47 phosphorylation in rat and porcine brain [44], and, most recently, T58 in rat heart [45].

## 4. Most Phosphorylations of Cyt*c* Are Protective through Partial Inhibition of ETC Flux and Decreased Apoptotic Activity

A general theme for phosphorylation sites identified on Cyt*c* is slowed electron transfer kinetics to COX and, although not universally true, decreased ability to activate downstream caspases. The binding of Cyt*c*, which is a highly basic protein with a pI of 9.6, with both COX and Apaf-1 is largely predicted to be driven by electrostatic interactions, with the positively charged Cyt*c* binding to negatively charged binding pockets on both proteins [2,17,52,53,54,55,56,57,58]. Therefore, it can be expected that phosphorylation, which introduces a negative charge, generally reduces the ability of Cyt*c* to interact with these two key partners.

Reducing ETC flux through phosphorylation of Cyt*c* may at first sight seem disadvantageous. However, similar to batteries that work best at an intermediate charge, the same argument can be made for mitochondria. This is because of the relationship between ETC activity, ΔΨ_m_, and mitochondrial ROS production [22,59,60]; mitochondria with optimal, intermediate ΔΨ_m_ levels in the range of about 100–130 mV generate ATP by ATP synthase at close to maximal capacity [61], whereas ROS production is minimal. At membrane potentials exceeding 140 mV, ROS production increases exponentially [62,63,64], which damages the cell and can result in cell death. Therefore, we propose that one major role of Cyt*c* phosphorylation is maintenance of optimal intermediate ΔΨ_m_ values, given that these phosphorylation sites tend to partially inhibit ETC activity, thus limiting mitochondrial ROS production to low basal levels. During acute conditions of stress, including ischemia as seen in ischemic stroke and myocardial infarction, these protective phosphorylations are lost in an attempt by the cell to boost ATP production and because kinases cannot operate due to ATP depletion while phosphatases are still operational or even activated due to mitochondrial calcium influx. Since there is no oxygen, the ETC is idling but primed for hyperactivity. During this time, succinate accumulates due to a reversal of complex II [65]. When blood flow is reinstated by mechanical or enzymatic removal of the clot, oxygen reenters the affected tissue. ETC activity resumes at pathological, maximal speed, leading to ΔΨ_m_ hyperpolarization and excessive ROS production at complexes I and III, which is further amplified by succinate-driven reverse electron transport to complex I (Figure 3) [65,66,67]. We propose that this sequence mechanistically explains reperfusion injury, which is a significant contributor to cell death on top of that caused initially by ischemia. Therefore, understanding the effect of post-translational modifications of Cyt*c* will provide a mechanistic understanding of pathophysiological processes that occur in a highly tissue-specific manner. Below we discuss in detail the functional effects of identified phosphorylations and acetylations of Cyt*c*.

### 4.1. Phosphorylation of Threonine 28 (T28)

T28 phosphorylation was mapped in Cyt*c* purified from bovine kidney [42]. For a post-translational modification to be biologically relevant, an appreciable portion of the protein pool must possess the modification. High resolution gel electrophoresis found that up to 83% of the protein pool purified from the entire kidney was phosphorylated. The V_max_ of COX in reaction with phosphorylated T28 Cyt*c* was 50% reduced compared to the maximal activity of COX in reaction with non-phosphorylated wild-type (WT) Cyt*c*. Additionally, the K_m_ decreased from 6.3 μM to 4.5 μM. Similar results were obtained using phosphomimetic T28E Cyt*c*, which showed a 73% reduction in V_max_ and a decrease in K_m_ from 7.8 μM to 4.7 μM in the reaction with COX [42]. These profound results may be explained by the strategic position of T28, which is part of the negative classical γ turn of Cyt*c,* which stabilizes the protein and is part of the positive epitope that facilitates Cyt*c* binding to its negatively charged binding site on COX [68]. Additionally, during the interaction between Cyt*c* and COX, T28 of Cyt*c* comes into close proximity with D50 and D51 of COX subunit I, which are known to be flexible residues, with D51 proposed to be involved in proton pumping [69,70]. Altogether, this suggests that phosphorylation of T28 perturbs the interaction between Cyt*c* and COX. Another study looked at T28D and found increased electron transfer to COX [46]. However, aspartate is not evolutionarily allowed at the site, while glutamate is allowed in certain species of plants, implying that T28E is the better phosphomimetic since it recapitulates the effects of phosphorylated Cyt*c* [71]. The opposing effects of T28E and T28D on COX activity may be explained based on structural considerations of the Cyt*c*-COX complex in which T28 localizes near K47 of COX subunit II [72]. Based on the charge–charge interaction model, both the T28E and T28D mutations should strengthen the interaction between Cyt*c* and COX, slowing down dissociation and thus the overall reaction. However, considering salt bridge formation, it is possible that electron transfer depends on the length and flexibility of the residue, which would be larger with T28E compared to T28D. With T28D, the complex could be restrained in a conformation allowing faster electron transfer, thereby explaining the opposite effects of the two mutations, despite showing the same charge change. Similar considerations may also apply to S47E and S47D (Figure 4).

The ability to activate downstream caspase-3 activity was unchanged comparing the phosphomimetic T28E to WT. The redox potential of T28E was decreased from 246 mV to 217 mV, which is near the typical reported literature range of 220 mV to 270 mV [73]. Compared to WT, T28E Cyt*c* showed an increased rate of reduction by ascorbate, sometimes considered a stand-in for Cyt*c* superoxide scavenging capability, and no change in rate of oxidation by H_2_O_2_. Interestingly, T28E Cyt*c* resisted degradation from high concentrations of H_2_O_2_ better than WT, tracked spectrophotometrically via decrease in heme absorbance. T28E also showed slightly reduced cardiolipin peroxidase activity [42].

The effects of T28 phosphorylation were also studied using Cyt*c* double knockout mouse lung fibroblasts, where both the somatic and testes isoforms of Cyt*c* are knocked out. Double knockout is necessary because knockout of just the somatic isoform induces expression of the testes isoform [74]. A construct for T28E was transfected into the double knockout cells. Clones expressing equal levels of WT or phosphomimetic T28E Cyt*c* were selected for further experimentation [42]. Cells expressing phosphomimetic T28E Cyt*c* showed no change in proliferation compared to cells expressing WT after trypan blue staining. It is known that mitochondrial respiration, ΔΨ_m_, and mitochondrial ROS production are linked [62]. This relationship was observed as the cells expressing phosphomimetic T28E showed 60% reduced intact cellular respiration, reduced ΔΨ_m_ measured via JC-1 fluorescence, 40% reduced mitochondrial ROS production measured via MitoSOX fluorescence, and 28% reduced ATP levels. Importantly, this demonstrates that modification of a single residue on Cyt*c* by phosphorylation or mutation can control electron flux through the entire ETC. Further supporting the protective role of this modification, cells expressing phosphomimetic T28E Cyt*c* underwent lower levels of cell death after treatment with low levels of H_2_O_2_ [42].

The signaling pathway for T28 phosphorylation was also elucidated. In vitro experiments with commercial AMPK and Cyt*c* revealed that AMPK selectively phosphorylated T28. AMPK and Cyt*c* were shown to interact via co-immunoprecipitation and co-localized together in the IMS via submitochondrial fractionation. Chemical activation of AMPK resulted in increased threonine phosphorylation of Cyt*c*, while pharmacological inhibition resulted in loss of threonine phosphorylation of Cyt*c*. Similarly, incubation of kidney tissue homogenates with pharmacological activators or inhibitors of AMPK resulted in increased or decreased respiration, respectively [42]. The kidney is an energetically demanding organ [75], and in contrast to other tissues, the kidney has high basal activity of AMPK and its dysregulation is known to result in kidney disease [76]. Furthermore, pharmacological activation of AMPK has been shown to be an effective treatment in a rat model of kidney ischemia-reperfusion injury [77].

Altogether, these experiments show that T28 phosphorylation is essential for kidney bioenergetic regulation. By partially inhibiting mitochondrial activity under basal conditions, T28 phosphorylation maintains an optimal intermediate ΔΨ_m_ and prevents the generation of excessive ROS. When this mechanism is lost, it would facilitate maximal respiration rates and result in increased ROS production [78].

### 4.2. Phosphorylation of Serine 47 (S47)

S47 phosphorylation was characterized in Cyt*c* purified from both pig and rat brain tissue and characterized [44]. S47 is the only serine residue present in porcine and rat somatic Cyt*c*; however, human Cyt*c* has a second serine residue at S15 [71]. Interestingly, S47 phosphorylation was found to be fully lost in ischemic brain tissue. Phos-tag gel electrophoresis determined that 35% of the pig Cyt*c* pool was phosphorylated, which probably is an underestimation because it takes time to open the skull and remove the brain, a period of time during which the brain is ischemic and likely starts losing the modification. Altered kinetics in reaction with COX were also observed when using S47 phosphorylated protein purified from control brain tissue. The activity of the Cyt*c*-COX reaction was reduced by 48% compared to the dephosphorylated protein purified from ischemic brain tissue. Docking simulations of the Cyt*c*-COX interaction predict that S47 of Cyt*c* would interact with K58 of COX subunit VIIa [52], highlighting the importance of this residue. Looking at the ability to initiate intrinsic apoptosis, S47 phosphorylated Cyt*c* showed 62% decreased caspase-3 activity compared to the dephosphorylated protein [44].

Using glutamate as a phosphomimetic replacement, activity of the Cyt*c*-COX reaction was 54% reduced and downstream caspase-3 activity was 65% reduced when S47E Cyt*c* was used compared to WT. These results are similar to those obtained using the in vivo phosphorylated protein purified from porcine brain, indicating that S47E reproduces functions of in vivo phosphorylation. Other findings using the S47E protein were that it was 33% more resistant to degradation at high concentrations of H_2_O_2_ and that the cardiolipin peroxidase activity was decreased [44].

Another study looked at phosphorylation via replacement with aspartate at this site [46]. Using this strategy, caspase-3 and cardiolipin peroxidase activities were decreased whereas Cyt*c*-COX activity was increased, which is opposite to the behavior of in vivo phosphorylated Cyt*c*. In addition, molecular dynamics simulations using the S47E crystal structure (6N10.pdb) demonstrate better spatial similarities between S47E and in silico phosphorylated S47 (Figure 4), implying that glutamate replacement is a better phosphomimetic replacement than aspartate [44].

The Cyt*c* knockout cell culture system was also employed to characterize the phosphomimetic S47E [47]. The cell line expressing phosphomimetic S47E Cyt*c* showed a 52% decrease in basal respiration with a concurrent increased extracellular acidification rate, a stand-in for glycolytic activity. As a result of the decreased ETC activity, cells expressing phosphomimetic S47E Cyt*c* also showed a decreased ΔΨ_m_, a 30% decrease in MitoSOX fluorescence indicating reduced mitochondrial ROS production, and a slight decrease in ATP levels. These experiments again highlight that a single modification or point mutation on Cyt*c* affects electron flux through the entire ETC and is able to induce global changes in overall mitochondrial activity. Using a model of oxygen-glucose deprivation followed by reoxygenation (OGD/R), which simulates ischemia-reperfusion injury in cell culture, the cells expressing WT showed a doubling in mitochondrial ROS production following OGD/R compared to control cells not subjected to OGD/R. In contrast, cells expressing phosphomimetic S47E Cyt*c* were unresponsive to OGD/R and did not show any differences in mitochondrial ROS production. Lastly, in agreement with data suggesting that phosphomimetic S47E replacement reduces the pro-apoptotic abilities of Cyt*c*, the cells expressing phosphomimetic S47E showed less cell death after exposure to H_2_O_2_ or OGD/R compared to WT [47].

Akt was identified as the putative kinase mediating S47 phosphorylation. In vitro kinase assays revealed that all three isoforms of Akt were able to phosphorylate Cyt*c* on S47 [47]. Akt, specifically Akt2, is known to translocate to the mitochondrial IMS [79,80,81,82]. Treatment of pig brain homogenates with activators or inhibitors of Akt induced or suppressed the phosphorylation of S47, respectively [47]. Altogether, these data show that S47 phosphorylation acts to maintain a healthy mitochondrial activity level and serves as a protective mechanism in brain tissue. Because this modification is lost during ischemia, and because the loss facilitates high rates of respiration during reperfusion, this causes ΔΨ_m_ hyperpolarization and thus mitochondrial ROS production [59], providing a molecular mechanism explaining, at least in part, reperfusion injury seen in brain tissue following stroke treatment [67].

### 4.3. Phosphorylation of Tyrosine 48 (Y48)

Cyt*c* purified from bovine liver was shown to be phosphorylated on Y48 [41]. The kinetics in reaction with COX were measured using the Y48 phosphorylated protein purified from bovine liver compared to protein dephosphorylated in vitro by shrimp alkaline phosphatase. The V_max_ of the Cyt*c*-COX reaction was 55% decreased compared to the dephosphorylated protein, and there was also a slight reduction in the K_m_ from 3.8 μM for dephosphorylated Cyt*c* to 3.0 μM for Y48 phosphorylated Cyt*c* [41]. Using a phosphomimetic glutamate replacement, Y48E Cyt*c* showed a similar change in V_max_, which was 30% decreased compared to WT [48]. Interestingly, phosphomimetic Y48E Cyt*c* showed an increase in K_m_ from 1.1 μM to 3.7 μM. The midpoint redox potential of the phosphomimetic Y48E Cyt*c* was greatly reduced to 192 mV in multiple publications [48,49], which is below the 220 mV to 270 mV range reported in the literature for Cyt*c* [73]. These results suggest significant perturbations to the heme crevice, altering the ability of the protein to carry out efficient electron transfer [49]. Furthermore, this redox potential value is below that of complex III, suggesting that electron transfer from complex III to Cyt*c* may also be impaired [83]. In addition, perturbing Y48 can be expected to disrupt this interaction because Y48 was predicted to be a key residue mediating the interaction between Cyt*c* and complex III [84].

Regarding pro-apoptotic functions, the phosphomimetic Y48E Cyt*c* was unable to activate downstream caspase-9 or caspase-3 activities in multiple studies [48,49,50]. Additionally, the cardiolipin peroxidase activity of phosphomimetic Y48E Cyt*c* was reduced, which can be explained by reduced binding to cardiolipin [48].

Y48 phosphorylation has also been studied using p-carboxymethyl-L-phenylalanine (pCMF) replacement, incorporated via the evolved tRNA synthetase method [85]. Work with the novel phosphomimetic Y48pCMF yielded some different results [50]. While there was agreement regarding decreased downstream caspase-3 activity, Y48pCMF showed increased activity with isolated COX and increased cardiolipin peroxidase activity. Interestingly, although Cyt*c*-COX activity was increased, the overall electron transfer rate through the ETC was decreased, possibly due to the perturbation between the Cyt*c*-complex III interaction.

Y48H is a known disease-causing mutation of Cyt*c* in humans, resulting in mild thrombocytopenia [86]. Y48H resulted in reduced respiration rate and increased apoptotic activity. Another study found that the mutation also caused the heme group to change to a pentacoordinated form, which increased cardiolipin peroxidase activity [87]. Given the crucial functions of Cyt*c* and its high evolutionary conservation, it is surprising that one of the few known disease-causing mutations in humans only causes mild thrombocytopenia. Perhaps only mild mutations of Cyt*c* are compatible with life.

### 4.4. Phosphorylation of Threonine 49 (T49)

T49 phosphorylation (numbering based on the mature protein, which lacks the start methionine) was reported to increase in aged mouse heart after detection via high-throughput phosphoproteomics [45]. It was characterized using the AC16 cell line, a human cardiomyocyte cell line, which was transfected using lentiviral constructs containing the sequence for phosphomimetic T49E Cyt*c*. Using this system, the authors reported lower levels of cell death and lower levels of caspase-9 and caspase-3 activities. However, the authors did not report knocking out endogenous Cyt*c* in their cell lines prior to transfection with the phosphomimetic T49E Cyt*c*, nor did they use the double Cyt*c* knockout mouse lung fibroblast cell line generated by the Moraes lab [74]. Therefore, the transfected cells likely express an unknown combination of both endogenous Cyt*c* in addition to phosphomimetic T49E Cyt*c* and data interpretation is thus rather limited.

### 4.5. Phosphorylation of Threonine 58 (T58)

T58 phosphorylation was mapped in Cyt*c* purified from rat kidney [43]. Unlike T28 phosphorylation, which was detected in all five kidney preparations tested [42], T58 phosphorylation was only detected in two of five kidney preparations [43]. T58 is known to be replaced with isoleucine in the testes isoform of Cyt*c* as well as in the single human Cyt*c* gene [71]. Using phosphomimetic T58E Cyt*c*, the kinetics of the Cyt*c*–COX reaction were interrogated. The V_max_ was 45% decreased compared to WT while the K_m_ was unchanged. This inhibition of Cyt*c*-COX activity is consistent with that seen with the other Cyt*c* phosphorylations and supports the model that Cyt*c* phosphorylation partially inhibits respiration to maintain optimal mitochondrial functioning [22,59].

Downstream caspase-3 activity of T58E Cyt*c* was 70% decreased compared to WT. The redox potential of T58E was reduced to 209 mV and the ROS scavenging ability of Cyt*c* was also altered; the T58E Cyt*c* demonstrated a 50% reduced rate of oxidation by H_2_O_2_ and a 50% increased rate of reduction by ascorbate. The heme group of the T58E Cyt*c* was slightly more resistant to degradation by high H_2_O_2_ concentrations compared to WT. Furthermore, the cardiolipin peroxidase activity of T58E Cyt*c* was reduced at high concentrations of cardiolipin. T58 has been reported to be part of the cardiolipin binding site on Cyt*c*, which suggests that the phosphorylation may partially disrupt this interaction [88].

Using the double Cyt*c* knockout system, cells transfected with T58E Cyt*c* showed a similar phenomenon to the other phosphorylations when expressed in cells: 68% decreased respiration rates in intact cells, reduced ΔΨ_m_ levels and mitochondrial ROS production, and 66% decreased ATP levels [43]. The cells expressing T58E Cyt*c* also underwent less cell death. Overall, these data establish the role of T58 phosphorylation as an anti-apoptotic modification, supporting the model that Cyt*c* phosphorylation sites maintain optimal mitochondrial functioning to limit ROS generation [22,59]. It still has to be shown in what renal cell type this modification is found and what kinase/phosphatase enzymes regulate this modification.

### 4.6. Phosphorylation of Tyrosine 97 (Y97)

Y97 was the first phosphorylation site of Cyt*c* to be discovered and characterized. It was initially mapped on Cyt*c* purified from bovine heart [40]. Using the in vivo phosphorylated purified protein, the K_m_ of the Cyt*c*-COX reaction was found to be increased from 2.5 μM of WT to 5.5 μM for the phosphorylated protein whereas the V_max_ was unchanged. This alteration of the Cyt*c*-COX kinetics is likely due to a perturbation of the heme group, which was seen on the UV–Vis spectra as the classic 695 nm peak, which assesses heme status via the M80-heme iron coordination, was shifted to 687 nm. Interestingly, another publication using phosphomimetic Y97E found no change of the redox potential and a significant decrease in melting temperature, implying that protein stability may be affected by alteration of this residue [49].

A third publication examined phosphorylation using phosphomimetic Y97pCMF replacement [51]. In contrast with the results obtained using the in vivo Y97 phosphorylated Cyt*c*, Y97pCMF demonstrated increased Cyt*c*-COX activity. The Y97pCMF did demonstrate some reduced pro-apoptotic functionality, with a decrease in downstream caspase-3 activity, although there was no change in cardiolipin binding or cardiolipin peroxidase activity using this phosphomimetic. Recently, a O-sulfotyrosine has become available as a novel mimetic, which should be tested as a model for studying tyrosine phosphorylation [89].

In a different study, Y97 phosphorylation was found to be rapidly induced by neuroprotective insulin treatment after induced ischemia-reperfusion in porcine and rat brains [90]. Induction of Y97 phosphorylation protected the CA-1 hippocampal neurons from cell death following the ischemia-reperfusion injury, supporting the anti-apoptotic role of Y97 phosphorylation seen using the phosphomimetic. Additionally, less Cyt*c* was released after 24 h post-ischemia when Y97 phosphorylation was induced by the insulin treatment, suggesting that phosphorylation may be able to prevent Cyt*c* release into the cytosol.

## 5. Characterized Acetylation Sites of Cyt*c*

In vitro lysine acetylation of Cyt*c* has been long studied. However, detection of lysine acetylation of Cyt*c* in vivo has only occurred recently and is a new area of research (Table 2). The first acetylation site discovered was lysine 8 (K8) in fasted mouse liver mitochondria via high-throughput acetyl-omics [91]. Later, lysine 53 (K53) was identified in prostate cancer [92]. This was followed by lysine 39 (K39) in ischemic skeletal muscle [93].

Lysine acetylation is the most abundant post-translational modification in mitochondria [94,95]. Lysine acetylation is believed to occur largely non-enzymatically in the mitochondrial matrix, due to the basic pH and high concentration of acetyl-CoA [96]. However, lysine acetylation of Cyt*c*, which resides in the IMS, was found to occur selectively on specific residues tied to a particular tissue or pathology. This suggests that specific acetyltransferase enzymes mediate these modifications. Unlike phosphorylations of Cyt*c*, which are present under basal conditions and are generally lost during ischemia, these acetylations tend to be associated with a specific pathophysiological state. Lysine acetylations suffer from the same methodological problems as phosphorylations: yields are typically low and the modification itself is transient. Therefore, it is common to study acetylation using glutamine as an acetylmimetic. In contrast to the positively charged side chain of lysine, acetyl-lysine and glutamine both have uncharged, polar side chains that contain amide groups [97]. Lysine acetylation of Cyt*c*, which leads to the loss of the positive charge of the ε-amine group, is generally predicted to interfere with binding to COX and Apaf-1, due to making Cyt*c* less positively charged. In this manner, acetylation, which removes a positive charge from a lysine residue, is similar to phosphorylation, which introduces a negative charge on a neutral S/T/Y residue, because both of them result in Cyt*c* becoming more negatively charged.

**Table 2 cells-13-00493-t002:** Summary of functionally characterized acetylation sites of Cyt*c*.

Residue	Tissue of Origin	Experimental Models	Findings
Lysine 8	Fasted Mouse Liver	Recombinant acetylmimetic K8Q Cyt*c*	Decreased Cyt*c*-COX activity, decreased K_D_ to cytochrome *c*_1_ [98]
Lysine 39	Ischemic Porcine Tibialis Anterior Muscle	In vivo acetylated Cyt*c* purified from ischemic porcine muscle	Increased Cyt*c*-COX V_max_, decreased caspase-3 activity [93]
		Recombinant acetylmimetic K39Q Cyt*c*	Increased Cyt*c*-COX V_max_, decreased caspase-3 activity, increased rate of oxidation, decreased rate of reduction, decreased cardiolipin peroxidase activity [93]
		Cyt*c* double knockout mouse lung fibroblasts expressing K39Q Cyt*c*	Increased respiration, increased ΔΨ_m_, increased mitochondrial ROS production, increased ATP levels, decreased cell death, reduced responsiveness to oxygen-glucose deprivation followed by reoxygenation [93]
Lysine 53	Human Prostate Cancer	Recombinant acetylmimetic K53Q Cyt*c*	Decreased Cyt*c*-COX V_max_, decreased caspase-3 activity, increased rate of oxidation, increased rate of reduction, increased heme degradation, decreased cardiolipin peroxidase activity [92,98]

### 5.1. Acetylation of Lysine 8 (K8)

K8 acetylation was discovered in a high-throughput study of fasted mouse liver [91]. This residue is known to interact with cytochrome *c*_1_ of Complex III [99,100], and it was found that acetylmimetic K8Q Cyt*c* had a perturbed dissociation constant from cytochrome *c*_1_ [98]. K8 of Cyt*c* is also known to interact with D139 of COX subunit II [72] as part of the proximal Cyt*c*-COX binding site, explaining the observed decrease in electron transfer to COX using acetylmimetic K8Q protein [98]. While this residue seems crucial for the electron transfer activity of Cyt*c*, not much is known regarding the effects of this modification on the other functions of Cyt*c*.

### 5.2. Acetylation of Lysine 39 (K39)

K39 acetylation was mapped in Cyt*c* purified from ischemic porcine tibialis anterior muscle after 45 min of ischemia [93]. The modification was not found in the muscle under basal conditions, making it the first reported gain of a post-translational modification during ischemia, suggesting that the response to ischemia may be muscle-specific. It is known that muscles tend to be more resilient to ischemia than other tissues such as brain, heart, kidneys, or liver [101,102]. K39 acetylation may provide a unique mechanism to potentially explain this resilience. Whereas other phosphorylations and acetylations of Cyt*c* tend to be present under basal conditions, lost during ischemia, and inhibitory to the Cyt*c*-COX reaction, K39 acetylation of tibialis anterior muscle follows a unique metabolic and apoptotic strategy by stimulating respiration to meet the cell’s energy demand and at the same time inhibiting apoptosis (Figure 5). K39 acetylation was found to be removed by sirtuin5, which is known to co-localize in the IMS with Cyt*c* [103,104]. Sirtuins are known to be downregulated in response to hypoxia, suggesting that preventing the removal of this acetylation during ischemia facilitates the protective effects [105]. The in vivo acetylated Cyt*c* purified from ischemic tibialis anterior muscle showed a 58% increase in Cyt*c*-COX V_max_, compared to the unacetylated Cyt*c* purified from non-ischemic tibialis anterior muscle [93]. Acetylmimetic K39Q Cyt*c* produced a similar increase of 38% compared to WT. Nuclear magnetic resonance studies demonstrated perturbation of residues interacting with or near the heme group such as the M80-containing loop and the K55-to-W59 stretch, possibly explaining the increased electron transfer rate by facilitating more efficient electron transfer. Interestingly, a separate publication studying K39L did not observe any changes in Cyt*c*-COX kinetics [106].

Additionally, this acetylation, which was triggered as a response to ischemia, blocks the pro-apoptotic functions of Cyt*c*, thereby acting as a protective mechanism during ischemia-reperfusion. Downstream caspase-3 activity using the in vivo acetylated protein purified from ischemic tibialis anterior muscle was 45% decreased, and the K39Q Cyt*c* showed a reduction of 90% in the same assay [93]. K39 of Cyt*c* interacts with the carbonyl carbon of F1063 of Apaf-1 [17,56], and the acetylation likely perturbs this interaction, decreasing downstream caspase-3 activity. Furthermore, cardiolipin peroxidase activity, another proapoptotic action, was decreased with the acetylmimetic K39Q Cyt*c*.

Interestingly, acetylmimetic K39Q Cyt*c* had a decreased rate of reduction by superoxide and an increased rate of oxidation by H_2_O_2_ compared to WT [93]. This may be because superoxide is a known signaling molecule produced during muscle contraction [107,108,109]. Therefore, reduced scavenging of superoxide could be an adaptative response to short periods of hypoxia that occur during normal muscle exercise, which becomes mal-adaptive in the context of ischemia-reperfusion injury. The crystal structure of the in vivo K39 acetylated Cyt*c* purified from ischemic tibialis anterior was solved at 1.5 Å and had an occupancy of 0.75 for the acetyl group, showing that a large proportion of the protein pool was acetylated following ischemia [93].

Similar to the results for the Cyt*c*-COX kinetics using purified proteins, cells expressing the acetylmimetic K39Q Cyt*c* showed an increase in respiration with a 70% increase in basal oxygen consumption rate. Accordingly, the basal ATP levels were increased 51% over WT. ΔΨ_m_ and mitochondrial ROS production also experienced increases compared to cells expressing WT. These experiments demonstrate that single point mutations on Cyt*c* can both inhibit and stimulate overall mitochondrial activity, supporting the concept that the Cyt*c*-COX interaction is the rate-limiting step of the ETC. After exposure to OGD/R, cells expressing K39Q Cyt*c* did not show a statistically significant increase in mitochondrial ROS production, unlike WT. These data show that K39Q Cyt*c* increases respiration, ΔΨ_m_, and ROS under baseline conditions compared to WT, but does not lead to a further increase following OGD/R, potentially protecting the cells from additional oxidative stress upon reperfusion. After exposure to H_2_O_2_, OGD/R, and thapsigargin treatment, cells expressing K39Q Cyt*c* experienced reduced levels of cell death. Using cells expressing K39Q Cyt*c*, reduced levels of Cyt*c* were co-immunoprecipitated with Apaf-1, suggesting impaired interaction of the two proteins. Furthermore, this translated into reduced cleavage of procaspase-9 and procaspase-3 after induction of apoptosis in the presence of K39Q Cyt*c*. Overall, skeletal muscle is a unique tissue type with drastic changes in perfusion depending on usage [110,111]. Cyt*c* K39 acetylation may be a skeletal muscles-specific adaptation during exercise or other short-term energy deficiencies, meant to allow the muscle to meet the high energy demand while protecting from cell death (Figure 5) [93].

### 5.3. Acetylation of Lysine 53 (K53)

Lastly, we will discuss K53 acetylation, which was mapped in Cyt*c* purified from human prostate cancer xenografts and prostate cancer specimens but was not found in normal prostate tissue [92]. Overall, this modification was found to be advantageous to cancer cells by driving two hallmarks of cancer: a metabolic switch from OxPhos to glycolysis and apoptosis evasion. It is known that acetyltransferase enzymes are upregulated in prostate cancer, which may help explain increased K53 acetylation [112,113]. Acetylmimetic K53Q Cyt*c* showed 35% decreased V_max_ for the Cyt*c*-COX reaction. K53 of Cyt*c* is within 5 Å of K58 residue of COX subunit VIIa [52], suggesting that K53 is within the Cyt*c*-COX binding domain and acetylation may spatially interfere with the interaction. This inhibition of respiration would assist the prostate cancer cells in metabolic reprogramming toward the glycolytic pathway.

Additionally, cancer cells often manage to evade apoptosis, which K53 acetylation facilitates. Downstream caspase-3 activity was 80% decreased using acetylmimetic K53Q Cyt*c* and may be explained because K53 is a known regulatory epitope for Apaf-1 binding [17]. The ability of acetylmimetic K53Q Cyt*c* to scavenge ROS was also significantly increased, with large increases in rate of oxidation by H_2_O_2_ and rate of reduction by superoxide, compared to WT. These may further facilitate prostate cancer cell survival by detoxifying ROS, blocking a strong pro-apoptotic signal. The pro-apoptotic cardiolipin peroxidase activity was also greatly reduced with acetylmimetic K53Q Cyt*c*. K53 is conserved in all mammals, further pointing to an important regulatory role of K53 for the interactions with both COX and Apaf-1 [71]. Future work should expand these studies to other cancers to see if this modification is more broadly involved in cancer pathology and potentially in resistance to cancer therapies relying on apoptosis induction.

## 6. Conclusions

The tissue and disease specific regulation of Cyt*c* has interesting implications. It has been long debated which step of the ETC is rate-limiting. We have previously proposed that the interaction between Cyt*c* and COX is rate-limiting [22]. Cyt*c* and COX are both highly regulated via allosteric ATP binding, post-translational modifications, and tissue-specific isoforms, and they are the only known mammalian components of the OxPhos that are regulated by all three mechanisms. Cyt*c* acts at the intersection point between pro-survival respiration and pro-death apoptosis, which depends on the subcellular localization of the protein, and, therefore, lends itself as an ideal regulatory target. Several reports employing traditional metabolic flux analysis experiments have identified COX as the rate-limiting step of the ETC [114,115,116,117,118]. For these experiments, an inhibitor of a specific complex is added in increasing amounts, and total flux through the ETC is measured, usually via oxygen consumption of COX as the terminal step of the ETC. The less inhibitors necessary to affect electron flux globally through the entire ETC, the more rate-limiting a particular step is. However, the possibility that Cyt*c* and its post-translational modifications are rate-limiting was overlooked in the past, perhaps due to the lack of a specific inhibitor of Cyt*c*.

Cyt*c*, which is crucial for physiological mitochondria functioning, has been shown to possess tissue-specific post-translational modifications that react to ischemia-reperfusion injury. Patterns within these post-translational modifications of Cyt*c* help explain different tissue sensitivities to ischemia-reperfusion injury. Specifically, various phosphorylations of Cyt*c*, which are present under basal conditions and lost during ischemia, sensitize the tissue to ischemia-reperfusion injury, while K39 acetylation of Cyt*c* in skeletal muscle, which is not present under basal conditions and is gained during ischemia, protects skeletal muscle and contributes to the resilience of this tissue type to ischemia-reperfusion injury. In this review, we have discussed that single post-translational modifications and point mutations of Cyt*c* are able to affect flux through the entire electron transport chain. When cells expressed T28E, S47E, T58E, or K39Q Cyt*c*, oxygen consumption rate by the cells was altered compared to cells expressing WT. While the phosphorylations of Cyt*c* and most acetylations are inhibitory to Cyt*c*-COX kinetics, K39 acetylation of Cyt*c* is stimulatory. This highlights that a change in Cyt*c*-COX activity, whether inhibitory or stimulatory, can decrease or increase total ETC flux, respectively.

This review has focused primarily on phosphorylations and acetylations of Cyt*c*, specifically those which have been detected in vivo in mammals and for which clear evidence exists that they are functionally important. Other post-translational modifications of Cyt*c* exist (reviewed in [23]). However, they were largely detected through high-throughput studies or are only found in vitro, therefore their biological significance is unknown. In most but not all cases, the eight modifications discussed here serve to support healthy cell functioning by blocking pro-apoptotic functions of Cyt*c* and by lowering mitochondrial activity to minimize ROS production. Furthermore, it has long been observed that different organs and even different tissue types within an organ have different sensitivities to ischemia and reperfusion injury [119]. For the phosphorylations, which are present in their respective tissues under basal conditions, their loss tends to drive ischemia-reperfusion injury by stimulating respiration, increasing ROS production, and sensitizing the cell to apoptosis. Overall, the phosphorylations occur in ischemia sensitive tissues, such as brain, heart, kidney, and liver, which may explain the susceptibility of these tissues to ischemia-reperfusion injury. K39 acetylation follows a different pattern, where acetylation is a response to ischemia, stimulating respiration and blocking apoptosis to enforce maximum muscle activity output. The presence of this acetylation in skeletal muscle, which is notoriously less sensitive to ischemic stress, results in skeletal muscle protection and resilience to ischemia-reperfusion injury. Altogether, the diversity of Cyt*c* regulation by post-translational modifications underscores the crucial nature of the protein for both the ETC and the cell as a whole.

## Figures and Tables

**Figure 1 cells-13-00493-f001:**
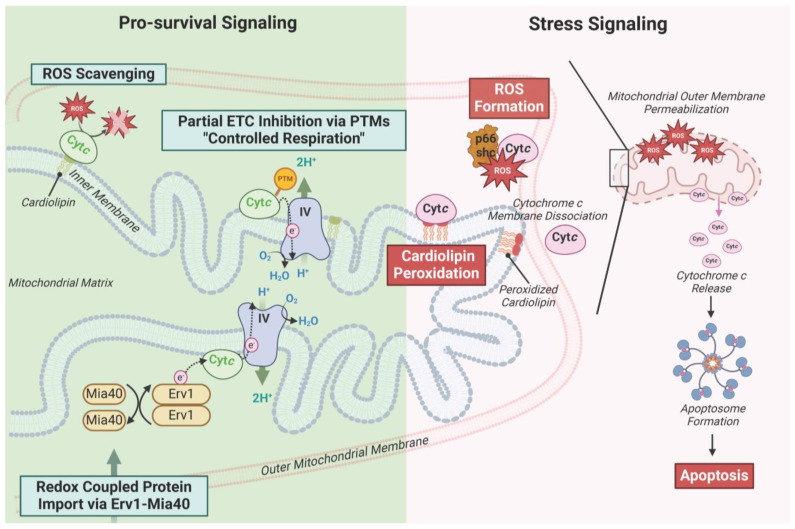
Functions of Cyt*c* divided into survival (left) and stress signaling functions (right). This figure was created using BioRender.com (accessed on 27 February 2024).

**Figure 2 cells-13-00493-f002:**
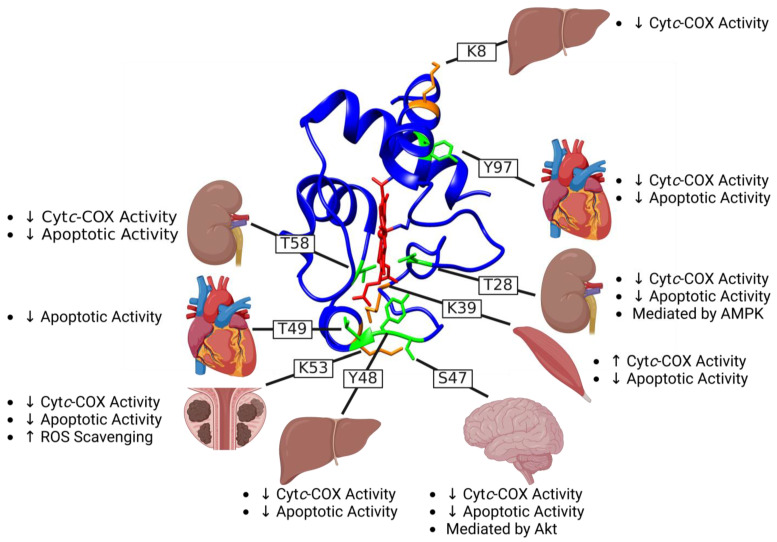
Tissue-specific phosphorylation (green) and acetylation (orange) sites of Cyt*c* mapped in mammalian tissues and their consensus functional effects. Discovery order: Y97 (heart), K8 (fasted liver), Y48 (liver), T28 (kidney), S47 (brain), T58 (kidney), K53 (prostate cancer xenografts), T49 (aged heart), K39 (ischemic tibialis anterior skeletal muscle). The heme group is annotated in red. The protein backbone is in blue except for serine/threonine/tyrosine residues that can be phosphorylated are in green and lysine residues that can be acetylated are in orange. The down arrow (↓) refers to decreased activity while the up arrow (↑) refers to increased activity. This figure of Cyt*c* (5c0z.pdb) was created using UCSF Chimera (v1.16) for protein visualization followed by BioRender.com (accessed on 27 February 2024).

**Figure 3 cells-13-00493-f003:**
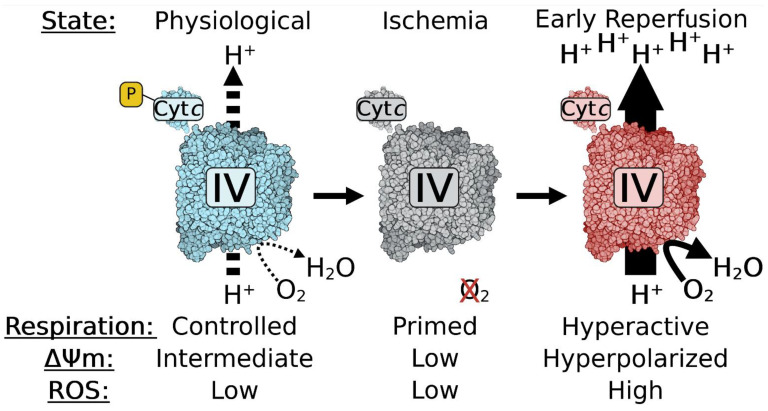
Inhibitory phosphorylations of Cyt*c* are lost during ischemia, promoting reperfusion injury. Inhibitory phosphorylations (yellow box) of Cyt*c* control respiration under physiological conditions (left, blue), which maintain an optimal intermediate ΔΨ_m_ and minimize ROS production. These partially inhibitory phosphorylations are lost during ischemia (middle, gray). Loss of phosphorylations primes the electron transport chain for hyperactivity, but lacks the terminal substrate, oxygen. When reperfusion occurs and oxygen is reintroduced into the tissue, the lack of inhibitory phosphorylations results in hyperactive mitochondria (right, red). This results in pathological high ΔΨ_m_ levels and bursts of ROS during reperfusion. Similar changes likely occur on COX and other components of the ETC. This figure was created using BioRender.com (accessed on 27 February 2024).

**Figure 4 cells-13-00493-f004:**
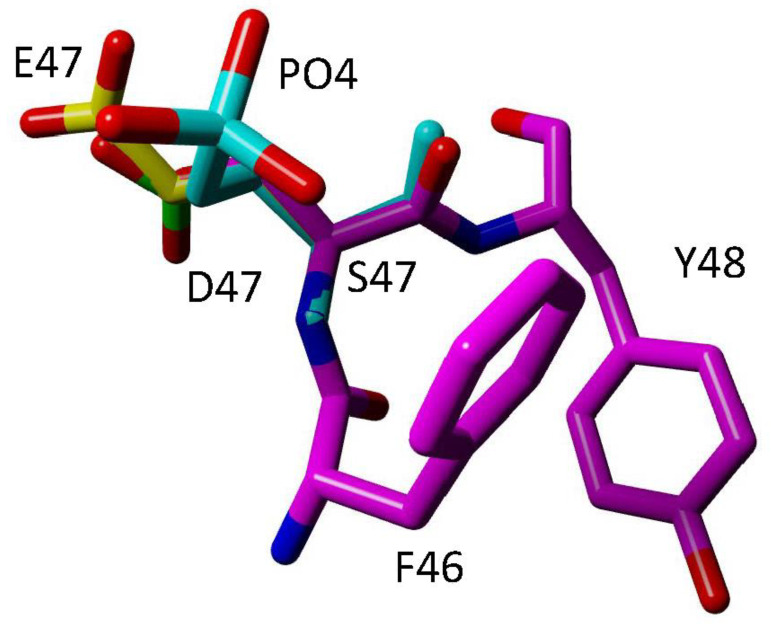
Comparison of phosphorylated S47, S47E, and S47D Cyt*c*. Overlay comparison shows that S47E (yellow) better approximates length of phosphorylated S47 (cyan) compared to the more spatially restricted S47D (green). The F46-S47-Y48 epitope (purple) of Cyt*c* is shown using pdb file 5C0Z. The S47E and S47D mutations were introduced with Yasara. The phosphorylation was introduced with COOT.

**Figure 5 cells-13-00493-f005:**
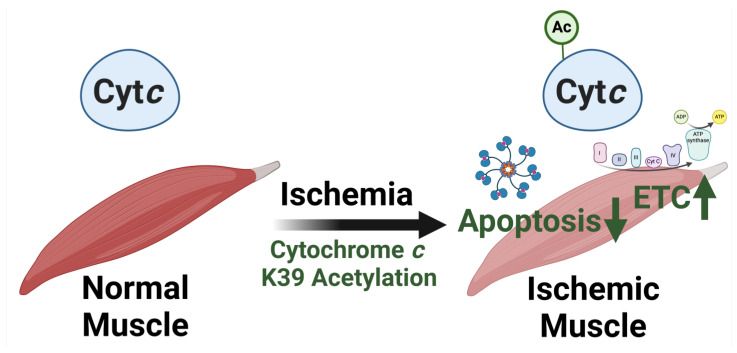
Tibialis anterior skeletal muscle Cyt*c* gains a post-translational modification during ischemia, activating electron transport chain flux and inhibiting apoptosis. In contrast to phosphorylations that are present under normal conditions but lost during ischemia in tissues such as brain, liver, kidney, and heart, Cyt*c* from tibialis anterior muscle becomes K39-acetylated during ischemia. This modification may explain resilience of skeletal muscle to ischemia-reperfusion injury because this acetylation inhibits the execution of apoptosis while activating respiration to quickly restore ATP production. This figure was created using BioRender.com (accessed on 27 February 2024).

**Table 1 cells-13-00493-t001:** Summary of functionally characterized phosphorylation sites of Cyt*c* in mammals. * Phosphomimetic replacement shows some opposite behavior from the phosphorylated protein.

Residue	Tissue of Origin	Experimental Models	Findings
Threonine 28	Bovine Kidney	In vivo phosphorylated Cyt*c* purified from bovine kidney	Decreased Cyt*c*-COX V_max_ and K_m_, phosphorylated by AMPK [42]
		Recombinant phosphomimetic T28E Cyt*c*	Decreased Cyt*c*-COX V_max_ and K_m_, decreased redox potential, increased rate of reduction, decreased degradation by H_2_O_2_, decreased cardiolipin peroxidase activity [42]
		Cyt*c* double knockout mouse lung fibroblasts expressing T28E Cyt*c*	Decreased respiration, decreased ΔΨ_m_, decreased mitochondrial ROS production, decreased ATP levels, decreased cell death [42]
		Recombinant phosphomimetic T28D Cyt*c*	* Increased Cyt*c*-COX activity, decreased redox potential, increased cardiolipin peroxidase activity [46]
Serine 47	Porcine Brain, Rat Brain	In vivo phosphorylated Cyt*c* purified from porcine brain	Decreased Cyt*c*-COX activity, decreased caspase-3 activity, phosphorylated by Akt [44,47]
		Recombinant phosphomimetic S47E Cyt*c*	Decreased Cyt*c*-COX activity, decreased caspase-3 activity, decreased heme degradation, decreased cardiolipin peroxidase activity [44]
		Cyt*c* double knockout mouse lung fibroblasts expressing S47E Cyt*c*	Decreased respiration, decreased ΔΨ_m_, decreased mitochondrial ROS production, decreased cell death, reduced responsiveness to oxygen-glucose deprivation followed by reoxygenation [47]
		Recombinant phosphomimetic S47D Cyt*c*	* Increased Cyt*c*-COX activity, decreased caspase-3 activity, decreased cardiolipin peroxidase activity [46]
Tyrosine 48	Bovine Liver	In vivo phosphorylated Cyt*c* purified from bovine liver	Decreased Cyt*c*-COX Vmax and K_m_ [41]
		Recombinant phosphomimetic Y48E Cyt*c*	Decreased Cyt*c*-COX Vmax, increased COX K_m_, decreased caspase-9 activity, decreased caspase-3 activity, decreased redox potential, decreased cardiolipin peroxidase activity [48,49]
		Recombinant phosphomimetic Y48pCMF Cyt*c*	* Overall decreased supercomplex activity (increased isolated Cyt*c*-COX activity), decreased caspase-3 activity, increased cardiolipin peroxidase activity [50]
Threonine 49	Mouse Heart	AC16 cardiomyocytes concurrently expressing WT Cyt*c* and T49E Cyt*c*	Decreased cell death, decreased caspase-9 activity, decreased caspase-3 activity [45]; methodological limitations are discussed in the text
Threonine 58	Rat Kidney	Recombinant phosphomimetic T58E Cyt*c*	Decreased Cyt*c*-COX V_max_, decreased caspase-3 activity, decreased rate of oxidation, increased rate of reduction, decreased heme degradation, decreased cardiolipin peroxidase activity [43]
		Cyt*c* double knockout mouse lung fibroblasts expressing T58E Cyt*c*	Decreased respiration, decreased ΔΨ_m_, decreased mitochondrial ROS production, decreased ATP levels, decreased cell death [43]
Tyrosine 97	Bovine Heart	In vivo phosphorylated Cyt*c* purified from bovine heart	Decreased Cyt*c*-COX K_m_, spectral shift of characteristic 695 nm peak to 687 nm (indicates changes to heme group) [40]
		Recombinant phosphomimetic Y97E Cyt*c*	Decreased melting temperature [49]
		Recombinant phosphomimetic Y97pCMF Cyt*c*	* Increased Cyt*c*-COX activity, decreased caspase-3 activity [51]

## Data Availability

Not applicable.

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
