# Peer review of "Phosphorylations and Acetylations of Cytochrome c Control Mitochondrial Respiration, Mitochondrial Membrane Potential, Energy, ROS, and Apoptosis"

_cells, 2024, doi:10.3390/cells13060493_

Round 1
Reviewer 1 Report
Comments and Suggestions for Authors
cells-2869671
This is an interesting and timely review, which reports a detailed analysis on the consequences of phosphorylations and acetylations on the multiple functions of cytochrome c. It will certainly be useful to the readers of Cells after some revision.
1. My main point is that the Authors should not forget other key mitochondrial events that impinge upon electron transfer at the cytochrome c junction, for example cristae rearrangement with widening of the crista junction by tBID [1], Ca2+-dependent swelling with mobilization of cytochrome c and decreased respiration [2], possible onset of the permeability transition with cytochrome c release [3,4], succinate accumulation and then triggering of reverse electron transfer with increased ROS production [5], oxidation of cardiolipin with further release of cytochrome c [6]. The implications of this omission are particularly evident in the Introduction, lines 60-62, “ …mitochondrial dysfunction triggers increased reactive oxygen species (ROS) production, which then initiates the release of Cytc into the cytosol” where the mechanistic link between increased ROS production and cytochrome c release remains otherwise very puzzling. This part of the Introduction is a convenient place for the integration of these and possibly additional points of regulation.
2. On the same note, lines 151-155 “When blood flow is reinstated by mechanical or enzymatic removal of the blood clot, oxygen reenters the affected tissue, and ETC activity resumes at pathological, maximal speed, leading to … hyperpolarization and excessive ROS production at complexes I and III …. We propose that this sequence mechanistically explains reperfusion injury,…” is way too simplistic. At least a mention to the role of Ca2+ deregulation and succinate accumulation during ischemia, matrix Ca2+ overload and reverse electron transfer at reperfusion, increased ROS and matrix swelling contributing to cytochrome c release is needed.
3. Abstract, “The phosphorylation sites are T28, S47, Y48, T49, T58, Y97 and tend to be present under basal conditions in a tissue-specific manner. In contrast, the acetylation sites are K8, 26 K39, and K53 and tend to be present in specific pathological conditions. All of the phosphorylation sites and two of the three acetylation sites partially inhibit respiration”. This should read “Phosphorylations at T28, S47, Y48, T49, T58, Y97 tend to be present under basal conditions in a tissue-specific manner. In contrast, acetylation at K8, 26 K39, and K53 tends to be present in specific pathological conditions. All phosphorylations and two of the three acetylations partially inhibit respiration” because the sites are always there, what matters is their modification.
4. Line 54 please remove “energy production” as energy cannot be produced! Perhaps “energy balance should be matched to demand”.
5. The single-letter code for aminoacids should not be listed as an abbreviation, please remove from the abbreviation list and use the single letter code for consistency in Figure 1, Tables 1 and 2, text at lines 110, 111, 254, 259, 260, 296, 342, 392, 397, 398, 400, 404, 407, 411, titles of paragraphs 4.1, 4.2, 4.3, 4.4, 4.5, 4.6, 5.1, 5.2, 5.3.
6. Line 308 caspase-9 is misspelled.
7. It is awkward to conclude a review with a Discussion paragraph, I suggest “Summary and conclusions” or something similar.
8. Line 515, remove “First being” as there is no second and third in the following sentences. Line 539, I would remove “In conclusion”, which would fit best the very last sentence of a manuscript. Note that a different font was used in lines 539-545.
References
1. Scorrano L, Ashiya M, Buttle K, Weiler S, Oakes SA, Mannella CA & Korsmeyer SJ (2002) A distinct pathway remodels mitochondrial cristae and mobilizes cytochrome c during apoptosis. Dev Cell 2, 55-67.
2. Bernardi P & Azzone GF (1981) Cytochrome c as an electron shuttle between the outer and inner mitochondrial membranes. J Biol Chem 256, 7187-7192.
3. Petronilli V, Nicolli A, Costantini P, Colonna R & Bernardi P (1994) Regulation of the permeability transition pore, a voltage-dependent mitochondrial channel inhibited by cyclosporin A. Biochim Biophys Acta 1187, 255-259.
4. Petronilli V, Penzo D, Scorrano L, Bernardi P & Di Lisa F (2001) The mitochondrial permeability transition, release of cytochrome c and cell death. Correlation with the duration of pore openings in situ. J Biol Chem 276, 12030-12034.
5. Chouchani ET, Pell VR, Gaude E, Aksentijevic D, Sundier SY, Robb EL, Logan A, Nadtochiy SM, Ord EN, Smith AC, Eyassu F, Shirley R, Hu CH, Dare AJ, James AM, Rogatti S, Hartley RC, Eaton S, Costa AS, Brookes PS, Davidson SM, Duchen MR, Saeb-Parsy K, Shattock MJ, Robinson AJ, Work LM, Frezza C, Krieg T & Murphy MP (2014) Ischaemic accumulation of succinate controls reperfusion injury through mitochondrial ROS. Nature 515, 431-435.
6. Ott M, Robertson JD, Gogvadze V, Zhivotovsky B & Orrenius S (2002) Cytochrome c release from mitochondria proceeds by a two-step process. Proc Natl Acad Sci U S A 99, 1259-1263.
Author Response
Please see the attached document for our response to the reviewer comments.

Reviewer 2 Report
Comments and Suggestions for Authors
Cytochrome c (Cytc) is a well-known mitochondrial protein located in mitochondrial intermembrane space, and its key function is to transport the electron between complex III and cytochrome oxidase. The cyt c is also known to trigger caspase-dependent apoptosis when it is released from mitochondria into the cytosol. The current review discussed the effect of post-translation modification on cyt c function. The authors acknowledged that phosphorylation of cyt c plays critical role in basal condition to optimize ATP production and decrease ROS generation. Acetylation may increase the resistance to ischemic damage, especially in skeletal muscle.
It is a well-written review paper. The reviewer has some minor comments.
Line 71 “Cytc scavenges and detoxifies ROS including superoxide and hydrogen per-oxide---"
Line 92 “Cytc catalyzes the formation of cardiolipin peroxide—"
Here authors talked about the two functions of cyt c.
One is antioxidant, and another is pro-oxidant function.
Could authors expand discussion here to talk about in what condition cyt c works as antioxidant, and in what other condition cyt c works as oxidant producer? Does posttranslational modification play the role in cyt c anti or prooxidant function?
These publications may need to be cited in the discussion PMID: 25092652 25092652
Line 154-155 “We propose that this sequence mechanistically explains reperfusion injury, which is a significant contributor to cell death on top of that caused initially by ischemia.—”
It is a good discussion about P-cyt c in ischemia-reperfusion injury. Could authors discuss the p-cyt c in other pathological conditions including aging and diabetic? It is well known that aging or diabetic impairs the COX that may keep the cyt c in relative reduced condition. Did P-cyt c play important role in reducing ROS generation in aging or diabetic conditions?
Line 224-225 “Furthermore, pharmacological activation of AMPK has been shown to be an effective treatment in a rat model of kidney ischemia-reperfusion injury [60]. “
Metformin is well known to activate AMPK and decrease ischemia-reperfusion injury. Does metformin treatment decrease kidney ischemia-reperfusion injury?
Line 391 “Characterized Acetylation Sites of Cytc”
Authors discussed several acetylation sites of cytc here. Could authors discuss the potential factors to induce cyt c acetylation including alteration of sirtuin1 or 3 ?
Author Response

(The authors gave the same response as above.)

Reviewer 3 Report
Comments and Suggestions for Authors
The authors review the dual roles of cytochrome c (Cyt c) in both promoting life through its function in the electron transport chain within mitochondria and facilitating cellular death by triggering apoptosis when released into the cytosol. They detail how activities of Cyt c are finely regulated by post-translational modifications, such as phosphorylation and acetylation, which are influenced by cellular conditions and stressors. Specifically, they identify six phosphorylation and three acetylation sites on Cyt c, noting how these modifications can regulate mitochondrial function to optimize membrane potential and minimize reactive oxygen species production, thus playing a crucial role in cellular response to stress, such as ischemia-reperfusion injury. The authors propose that these regulatory mechanisms underline critical importance of Cyt c in managing mitochondrial function, apoptosis, and overall cellular health.
Although this review is well written, two important issues are not discussed in it: the role of cytochrome c bound to the mitochondrial membrane/free state and its role in pathological conditions such as ischemia.
Second - the role of oxidation/reduction of free cytosolic cytochrome c in the pathologic processes in the cell.
Please consider discussing these papers:
doi: 10.1016/j.phrs.2016.03.036
doi: 10.1016/j.biocel.2012.07.022
Also, in Lines 539-545 please check Font.
Addition of a Figure summarizing most important biological processes where cytochrome c participates, would be very useful for better understanding.
Author Response

(The authors gave the same response as above.)

Reviewer 4 Report
Comments and Suggestions for Authors
This manuscript by Hüttemann and co-workers offers an excellent review on the effect of posttranslational modifications (PTMs) affecting cytochrome c (Cyt-c) on ROS production, efficiency of oxidative phosphorylation —focusing specially on COX activity— and cell death and survival pathways. Authors make a thorough but fair criticism of the different models and approaches to the distinct features affected by PTMs, highlighting consistencies much more than discrepancies, which are still mentioned. The result is a compact and consistent model explaining how absence and presence of different PTMs at distinct Cyt-c residues determine resilience or sensitivity of tissues towards stress. This excellent manuscript is easy to read an enjoyable. Consequently, I fully recommend it for publication as it is. Nevertheless I find there are several points that authors could find worth discussing more.
1) The first one concerns the differences between the distinct residues used to engineer fosfomimic Cyt-c. First, authors inticate at the beginning of the manuscript (line 133) that posphorylation mainly provides a shift of cyt-c net charge, ant that could shallowy explain part of the observed effects. For this being totally true, the altered charges need to be free of screening by surrounding ionizable groups, as it is the electrostatic potential, rather than the net charge.
Then, residue volume —length?— becomes important: the replacements of hydroxyl-bearing residues by aspartate and glutamate cause distinct effects. Authors propose the single ethylene difference between the two residues being key to determine the formation of salt bridges with COX. Authors justify this statement with docking computations showing Cyt-c T28 locating in the proximity ofo COX D51. However, the structure of the COX:Cyt-c complex has been solved later by X-ray Difraction (XRD) at 2 Å resolution (ref. 88 in the manuscript). In this structure (pdb ID: 5IY5), T28 falls near a lysine residue of COX surface (K47). According to this using the charge-charge interaction model as proposed, both the T28E and the T28D mutations might increase, not decrease, COX binding by Cyt-c. However, considering this salt bridge as a lead, however, it is possible that the electron transfer would depend in its length and flexibility, which should be larger when Thr28 is replaced by glutamate. When Thr28 is replacedby aspartate, the complex could be restrained in a less efficient conformation, thereby explaining the opposite effects of the two mutations, despite showing the same charge shift. Same applies to S47D/E. In addition the interactions with solvent —according to reference 88, the electron transfer pathway for the WT is particularly solvated with an ordered water network of — could become relevant in this context.
Siimilarly, the observations made after tyrosine substitutions by glutamate and p-carboxymethyl-phenylalanine are different, and both asre somewhat different to what is observed with the truly phosphorylated protein. This is particularly true for Y97E, and also apllies to Y97pCMF. Phosphotyrosine has a larger volume, and phosphate can also establish a larger amount of hydrogen bonds, being also highly polarizable.As Tyr97 is mostly buried inside Cyt-c, volume and branching becomes important, in addition to formal charges. At this point it is worth to mention that novel residues such as O-sulfotyrosine (sTyr)1 have become available recently. Maybe authors could make a comment on their putative usefulness on this topic.
2) A second topic for discussion is that authors highlight COX activity being limiting one in the electron tranport chain. Then, as expected and in agreement with author’s explanations, it seems that Vmax for COX reduction can be the parameter to look at, since it is the common feature in most in vitro analyses, whereas the effect of phosphomimics on the Michaelis constant (KM) seems to vary depending on the phosphomimics. Part of the changes in the KM values could be of course attributed to changes in the catalytic constant (directly related to Vmax), but maybe part of them are also due to changes in the affinity or the lifetime of the complex. In this case, the turnover of reduced donor would be affected, in addtion to Vmax. There is at least an example in the literature in which increasing Vmax many folds but affecting turnover can fully disrupt the whole electron transport chain. This could explain numerical differences between the studies performed in vitro and in cells. A very brief comment on this point would be thanked.
3) Lines 254-260. At this point, a figure with an structural overlay of S47D and S47E would be interesting.
4) Although the manuscript focuses on phosphorylation and acetylation, Cyt-c can undergo other PTMs that could affect its functionality —namely nitration and homocysteinylation, the latter only observed in vitro —. Maybe the authors would like to make any comment about possible similarities or differences with respect to the PTMs being the topic of the manuscript.
5) Lines 406-408. Similar to what was discussed in (1). Lysine acetylation causes a change in the net charge of the protein, but it is also true that acetyllysine maybe also somewhat more hydrophobic than lysine. Please extend a bit the description of the modified residue properties.
6 ) Text in lines 234 to 236 seems somehwhat confusing.
1. Italia, J. S.; Peeler, J. C.; Hillenbrand, C. M.; Latour, C.; Weerapana, E.; Chatterjee, A., Genetically encoded protein sulfation in mammalian cells. Nat. Chem. Biol. 2020, 16 (4), 379-382.
Author Response

(The authors gave the same response as above.)
